



Title
Bias correction of gauge-based gridded product to improve extreme precipitation analysis in the
Yarlung Tsangpo-Brahmaputra River Basin
Author names and affiliations
Xian Luo[1,2], Xuemei Fan[1], Yungang Li[1,2], and Xuan Ji[1,2]
[1]Institute of International Rivers and Eco-security, Yunnan University, Kunming, China
[2]Yunnan Key Laboratory of International Rivers and Transboundary Eco-security, Kunming, China
Address
Institute of International Rivers and Eco-security, Yunnan University,
South Section, East Outer Ring Road, Chenggong District, Kunming, China
Email
Xian Luo: luoxian@ynu.edu.cn
Xuemei Fan: fanxuemei7@163.com
Yungang Li: ygli@ynu.edu.cn
Xuan Ji: jixuan@ynu.edu.cn
Contact Author: Xian Luo (luoxian@ynu.edu.cn)
Second Contact Author: Yungang Li (ygli@ynu.edu.cn)



**Abstract**. Critical gaps in the amount, quality, consistency, availability, and spatial distribution of
rainfall data limit extreme precipitation analysis, and the application of gridded precipitation data
are challenging because of their considerable biases. This study corrected Asian Precipitation Highly
Resolved Observational Data Integration Towards Evaluation of Water Resources (APHRODITE)
in the Yarlung Tsangpo-Brahmaputra River Basin (YBRB) using two linear and two nonlinear
methods, and assessed their influence on extreme precipitation indices. The results showed that the
original APHRODITE data tended to underestimate precipitation during the summer monsoon
season, especially in the topographically complex Himalayan belt. Bias correction using
complementary rainfall observations to add spatial coverage in data-sparse regions greatly improved
the performance of extreme precipitation analysis. Although all methods could correct mean
precipitation, their ability to correct the wet-day frequency and coefficient of variation were
substantially different, leading to considerable differences in extreme precipitation indices.
Generally, higher-skill bias-corrected APHRODITE data are expected to perform better than those
corrected by lower-skill approaches. This study would provide reference for using gridded
precipitation data in extreme precipitation analysis and selecting bias-corrected method for rainfall
products in data-sparse regions.

**1   Introduction**
Extreme precipitation often leads to floods, debris flows, and other secondary disasters (Wang
et al., 2017), and changes in the frequency and intensity of extreme precipitation profoundly
influence both natural environment and human society profoundly (Easterling et al., 2000; Yucel
and Onen, 2014). Rainfall observations provide a primary foundation for comprehending their long-



term variability and change in extreme precipitation (Alexander, 2016). Accurate rainfall data are
necessary for flood protection and water resource management. However, due to scarce spatial
coverage of rainfall stations, short-length rainfall records, and high proportions of missing data,
observations currently available in some remote basins are clearly inadequate to capture their
precipitation characteristics. In addition, observed rainfall data are usually difficult to collect in
international river basins because many countries may not share or freely distribute data (Lakshmi
et al., 2018).

The Yarlung Tsangpo-Brahmaputra River is the fourth largest river in the world in terms of

flow (Kamal-Heikman et al., 2007), which is influenced profoundly by complex atmospheric
dynamics and regional climate processes (Immerzeel et al., 2010; Pervez and Henebry, 2015).
Because its agriculture and economy rely heavily on monsoon precipitation, the basin is particularly
vulnerable to changing climate (Singh et al., 2016; Liu et al., 2018; Janes et al., 2019; Xu et al.,
2019; Zhang et al., 2019). During the four summer monsoon months of June, July, August, and
September (JJAS), extreme precipitation with large uncertainties leads to numerous floods (Kamal-
Heikman et al., 2007; Dimri et al., 2016; Malik et al., 2016). However, the understanding on extreme
precipitation in the Yarlung Tsangpo-Brahmaputra River Basin (YBRB) have a number of gaps
because of its complex topographic interactions with atmospheric flows, lack of observations, and
data sharing issues, which hinder effective flood management (Ray et al., 2015; Prakash et al., 2019).

Currently, different gridded rainfall products provide effective information over regional to

global scales, which could be broadly classified into four categories: (1) gauge-based data sets that
build on observations from rainfall stations; (2) products from numerical weather predictions or
atmospheric models; (3) satellite-only products; and (4) combined satellite-gauge products. The



performance of these products varies from region to region (Duan et al., 2016). Given the
heterogeneity of orography and climate in the YBRB, observing and modeling its precipitation are
very challenging (Khandu et al., 2017). In addition, satellite products are less reliable because high
convective rainfall generally takes place in the southern foothills of the Himalayas (Prakash et al.,
2015). Compared with some other gauge-based products, the Asian Precipitation Highly Resolved
Observational Data Integration Towards Evaluation of Water Resources (APHRODITE) dataset
collected more rainfall observations across South Asia (Rana et al., 2015), which have been proved
could better estimate spatial precipitation (Andermann et al., 2011). Nonetheless, the lack and
uneven distribution of rainfall stations at high altitudes in the Tibetan Plateau and Himalayas may
introduce uncertainty and affect the accuracy of APHRODITE (Rana et al., 2015; Chaudhary et al.,

2017).

Numerous rainfall observations can be obtained from public databases, although their short

record and static character limit their direct application in precipitation analysis (Donat et al., 2013).
However, these data could be useful for bias correction of gauge-based gridded products by
providing additional observations from the denser network of rainfall stations. On the other hand,
ranging from simple linear scaling to more sophisticated nonlinear approaches, several methods
have been developed to adjust global climate model (GCM) data (Teutschbein and Seibert, 2012).
Similarly, these bias correction methods could be applied to correct gridded rainfall products in
sparsely-gauged mountainous basins (He et al., 2017). It is important to study whether extreme
precipitation analysis could be improved by bias correction of gridded precipitation data and how
different methods would influence extreme precipitation indices.

This study evaluated different bias correction approaches for APHRODITE in the YBRB and





assessed their effects on extreme precipitation analysis. We first corrected APHRODITE dataset by
both linear and nonlinear methods, and then evaluated their performances. Next, we calculated
extreme precipitation indices using the original and different corrected APHRODITE to further
investigate the effects of bias correction on extreme precipitation analysis. The results would support
reference for the application of gridded precipitation data and bias-corrected methods in extreme
precipitation analysis.

**2   Material and methods**
**2.1   Study area**
The YBRB can be divided into three physiographic zones: (1) the Tibetan plateau (TP),
covering 44.4% of the basin, with elevations above 3500 m; (2) the Himalayan belt (HB), accounting
for 28.6% of the basin, with elevations ranging from 100 m to 3500 m; and (3) the floodplains (FP),
covering 27.0% of the basin, with elevations up to 100 m (Immerzeel, 2008).
The moisture in the YBRB is mainly from the Indian Ocean. The YBRB exhibits a broad range
of precipitation from the semi-arid upstream areas to the HB characterized by abundant orographic
rainfall as well as the vast humid FP. In the upstream areas, precipitation is concentrated during
JJAS, and rainfall intensity is mostly low due to long-distance moisture transport (Guan et al., 1984).
The irregular topographic variations in the Himalayas profoundly affect the spatial distribution of
precipitation by altering monsoonal flow, producing intense orographic rainfall along the Himalayan
foothills (Khandu et al., 2017). The downstream areas also receive high rainfall from monsoon flow
during JJAS, accounting for 60%−70% of the annual rainfall (Gain et al., 2011).

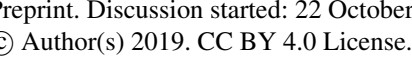

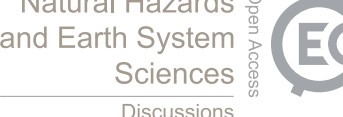
### 2.2 Data sources

#### 2.2.1 Observational data

In the upper YBRB, rainfall data across China recorded at 31 meteorological stations were collected from the National Meteorological Information Center (NMIC, sourced from the China Meteorological Data Sharing Service System). In addition, data observed at 91 rainfall stations in the downstream area were obtained from the Global Historical Climatology Network (GHCN)−Daily for bias correction. GHCN-Daily comprises observations from four sources, which have been undergone extensive quality reviews, including the U.S. Collection, the International Collection, the Government Exchange Data, and the Global Summary of the Day. The locations of rainfall stations are shown in Fig. 1.

#### 2.2.2 APHRODITE

Numerous rainfall observations were incorporated into APHRODITE, including (1) Global Telecommunication System (GTS)-based data, (2) data obtained from other projects or organizations, and (3) their own collection. The ratios of rainfall observations after quality control to the world climatology were calculated and interpolated for each month. The interpolated ratios were multiplied by the world climatology, and the first six components of the fast Fourier transform of the resulting values were used to obtain daily precipitation (Yatagai et al., 2012).

Daily rainfall data of APHRO_MA_025deg_V1101 (http://aphrodite.st.hirosaki-u.ac.jp/index.html) at 0.25° resolution in the Asian monsoon area end in 2007, while recently published APHRO_MA_025deg_V1101EX_R1 (http://aphrodite.st.hirosaki-u.ac.jp/index.html), using the same algorithm and spatial resolution, extend the time series over the period 2007−2015.




Therefore, extreme precipitation could be analyzed during 1951−2015 by applying both datasets.
To investigate the influence of topography on bias-corrected APHRODITE, the APHRODITE grids
were classified into three topographic zones (the TP, HB, and FP; Fig. 2).

**2.3    Methods**
**2.3.1    Bias correction methods**

Two linear methods (linear scaling (LS) and local intensity scaling (LOCI)) and two non-linear

methods (power transformation (PT) and quantile−quantile mapping (QM)) were used for bias
correction in this study.
(1) LS

LS corrects monthly estimates in accordance with observations (Lenderink et al., 2007). It

adjusts APHRODITE using the ratio between mean monthly observations and corresponding
estimations:
$$P^*_{APH}(d) = P_{APH}(d) \cdot \left[ \frac{\mu_m(P_{obs}(d))}{\mu_m(P_{APH}(d))} \right]$$    (1)
where $P_{APH}(d)$ and $P^*_{APH}(d)$ are the original and corrected APHRODITE, respectively.
$\mu_m(P_{obs}(d))$ and $\mu_m(P_{APH}(d))$ are the mean monthly observation and corresponding
APHRODITE, respectively.
(2) LOCI

LOCI makes a flexible adjustment to the wet-day frequency and intensity (Schmidli et al., 2006;

Teutschbein and Seibert, 2012). Firstly, an adjusted precipitation threshold ($P_{th,APH}$) is determined
so that the threshold exceedance matches the wet-day frequency of the observation. Secondly, a
linear scaling factor for wet days is computed, using the mean monthly precipitation:



$$s = \frac{\mu_m\left(P_{obs}(d)\middle|P_{obs}(d) > 0 \ \mathrm{mm}\right)}{\mu_m\left(P_{APH}(d)\middle|P_{APH}(d) > P_{th,APH}\right) - P_{th,APH}} \qquad (2)$$

Finally, the precipitation data are corrected, using:
$P_{APH}^{*}(d) = \max\left(s \cdot \left(P_{APH}(d) - P_{th,APH}\right), 0\right)$    (3)
(3) PT

PT corrects both the mean and the coefficient of variation of precipitation (Leander and

Buishand, 2007), changing precipitation by:
$P_{APH}^{*}(d) = a \cdot \left(P_{APH}(d)\right)^{b}$    (4)
where $a$ and $b$ are the parameters of the power transformation, which are obtained using a
distribution-free approach and estimated for each month within a 90-day window. Using a root-
finding algorithm, the value of $b$ is firstly determined to ensure that the coefficient of variation of
the corrected precipitation matches that of the observed precipitation. The parameter $a$ is then
calculated using the mean observation and the corresponding mean of the transformed values.
(4) QM

By shifting occurrence distributions, QM corrects the distribution function of the APHRODITE

to match those of the observed distribution function. A Gamma distribution is usually assumed for
precipitation events and has been proven to be effective in precipitation analysis (Teutschbein and
Seibert, 2012):
$f_{\gamma}\left(x\middle|\alpha,\beta\right) = x^{\alpha-1} \cdot \dfrac{1}{\beta^{\alpha} \cdot \Gamma(\alpha)} \cdot e^{-\frac{x}{\beta}}; x \geq 0; \alpha, \beta > 0$    (5)
where $\alpha$ and $\beta$ are the shape parameter and scale parameter, respectively.

The cumulative density function (CDF) for the APHRODITE is matched with that for the daily

observed precipitation for a given month, and the daily precipitation for APHRODITE is corrected
depending on its quantile. It should be noted that for APHRODITE, many days had low precipitation





estimates instead of substantial dry conditions, which may distort the distribution of daily
precipitation. Therefore, an adjusted precipitation threshold is also used to ensure the wet-day
frequency of the corrected APHRODITE match the observed frequency:
$$P_{APH}^{*}\left(d\right)=\begin{cases}0, & \text{if} \quad P_{APH}\left(d\right)<P_{th,APH}\\ F_{\gamma}^{-1}\left(F_{\gamma}\left(P_{APH}\left(d\right)\middle|\alpha_{APH,m},\beta_{APH,m}\right)\middle|\alpha_{obs,m},\beta_{obs,m}\right), & \text{otherwise}\end{cases}$$
(6)

$F_{\gamma}$  and  $F_{\gamma}^{-1}$  are the Gamma CDF and its inverse, respectively.

Hereafter, the APHRODITE data corrected by LS, LOCI, PT, and QM are referred as LS-

APHRODITE, LOCI-APHRODITE, PT-APHRODITE, and QM-APHRODITE, respectively.

**2.3.2   Evaluation of APHRODITE estimates**

Observed data from rainfall stations were applied to evaluate the performances of the original

and corrected APHRODITE at daily scale. Five common statistical metrics, including Pearson
correlation coefficient (*r*), percentage bias (*PB*), mean error (*ME*), mean absolute error (*MAE*), and
root mean squared error (*RMSE*), were calculated (Duan et al., 2016), and their equations and
optimal values are summarized in Table 1.

**2.3.3   Indices of extreme precipitation**

To characterize extreme precipitation during JJAS, six indices recommended by the Expert

Team on Climate Change Detection and Indices (ETCCDI), including consecutive wet days (CWD),
number of heavy precipitation days (R10mm), number of very heavy precipitation days (R20mm),
maximum 1-day precipitation amount (Rx1d), maximum 5-day precipitation amount (Rx5d), and
simple daily intensity index (SDII), were applied in this study. Detailed descriptions of these indices
are shown in Table 2. The indices fall roughly into three categories: (1) duration indices, which



represent the length of the wet spell; (2) threshold indices, which count the days on which a fixed
precipitation threshold is exceeded; (3) absolute indices, which describe the maximum 1-day or 5-
day precipitation amount (Sillmann et al., 2013).
In the grids distributed with rainfall stations, these six indices were calculated from the
corrected APHRODITE. In addition, spatial interpolation was performed using inverse distance
weighted (IDW) to obtain extreme precipitation indices for other grids within the basin. This
allowed us to calculate mean values for each of the three topographic zones.

**3 Results**
**3.1 Evaluation of original and corrected APHRODITE estimates**
The statistical metrics for daily precipitation during JJAS calculated for both original and
corrected APHRODITE are summarized in Table 3. In general, original APHRODITE estimated
precipitation well during JJAS in the YBRB, yielding $r$ close to 0.8 in all three zones. However, the
$PB$ of the original APHRODITE estimates in the TP, HB, and FP were −9.4, −24.2, and −11.4,
respectively. This indicates that they tended to underestimate precipitation. Due to the high
orographic precipitation coupled with the low density of rainfall stations used in the APHRODITE,
underestimation in the HB with complex topography was greatest.
Corrected APHRODITE estimates yielded better statistical metrics. The $PB$ and $ME$ for LS-,
LOCI-, and PT-APHRODITE were almost 0, indicating there was no longer any distinct bias in the
mean of daily precipitation. The linear approaches and PT calculate correction values based on the
ratio between long-term observations and APHRODITE estimates. Therefore, LS-, LOCI-, and PT-
APHRODITE agreed with the mean observations. In the case of QM-APHRODITE, the $PB$ in the



TP, HB, and FB were 3.2, 11.3, and 5.7, respectively, which were larger than those for other
corrected APHRODITE estimates.

The other three statistical metrics ($r$, $MAE$, and $RMSE$) for the corrected APHRODITE in the

TP were similar to those for the original APHRODITE, while the corrected APHRODITE in the FP
had slightly higher $r$ and lower $MAE$ and $RMSE$. In the HB, the $r$, $MAE$, and $RMSE$ for the original
APHRODITE were 0.81, 3.6 mm, and 15.9 mm, respectively; while for the corrected APHRODITE,
the $r$ were all higher than 0.9, and the $MAE$ and $RMSE$ were mostly less than 3 mm and 10 mm,
respectively, suggesting that the greatest improvement occurred in the HB.

**3.2   The influence of bias correction on extreme precipitation indices**
**3.2.1   Spatial distribution of extreme precipitation**

Rainstorms over the lower YBRB usually have a duration of 2−3 days (Dhar and Nandargi,

2000), and large multi-day precipitation events are crucial to the floods in the basin. Hence, the
spatial distribution of Rx5d during JJAS based on the original APHRODITE estimates were
compared with the corrected APHRODITE estimates in Fig. 3. For the original APHRODITE, the
area with Rx5d higher than 300 mm only accounted for 2.0% of the basin, while the proportions for
LS-, LOCI-, PT-, and QM-APHRODITE were 10.9%, 18.7%, 21.7%, and 21.3%, respectively. The
most profound difference between the original and corrected APHRODITE occurred over the
windward slopes of the Himalayas before the river flows into the Brahmaputra valley. The Rx5d
calculated from the original APHRODITE were lower than 300 mm, while much higher Rx5d were
obtained after bias correction, yielding maxima of 946.6, 1030.3, 1105.1, and 1396.6 mm for LS-,
LOCI-, PT-, and QM-APHRODITE, respectively. The eastern Himalayas, acting as orographic



barriers, push the southwest moist air upwards, leading to heavier extreme precipitation over the
windward slopes (Singh et al., 2004; Bookhagen and Burbank, 2010; Dimri et al., 2016). However,
original APHRODITE estimates tended to substantially underestimate these extreme precipitation,
likely because of sparse rainfall gauge data. Besides aforementioned region, higher Rx5d along the
Himalayan front were also found after bias correction. In this case, extreme precipitation calculated
from nonlinear approaches were heavier than those derived from linear methods. Bias correction are
able to consider topographic effects on the spatial distribution of extreme precipitation more
comprehensively by making use of observations from denser network of rainfall stations. This
resulted in better capturing of the main climatological features of extreme precipitation in the YBRB.

**3.2.2    Extreme precipitation indices in the three physiographic zones**
Sparsely distributed rainfall stations and short records affect the accuracy of spatial
precipitation interpolation. Hence, it is hard to directly evaluate extreme precipitation obtained from
bias-corrected APHRODITE by carrying out pixel-to-pixel comparison with those interpolated
using gauge observations. A major limitation is the remaining uncertainty regarding how well
different corrected APHRODITE estimate heavy rainfall, especially in data-sparse regions. Despite
improved statistical metrics for bias-corrected APHRODITE, these could not guarantee good
performance in extreme precipitation analysis. To obtain valuable information about the influences
of bias-corrected methods on extreme precipitation analysis, extreme precipitation indices
calculated from the original and four corrected APHRODITE were compared.
Extreme precipitation indices calculated from the original and four corrected APHRODITE
estimates in the three different physiographic zones are shown in Fig. 4. The CWD estimated using



original APHRODITE and LS-APHRODITE were similar. Meanwhile, those derived from LOCI-,
PT-, and QM-APHRODITE estimates were much less. For the original APHRODITE, there were a
lot of days with low precipitation estimations instead of substantial dry conditions, leading to the
overestimation on CWD. Likewise, this propagated to the LS-APHRODITE, because there was no
change made to the wet-day frequency. In contrast, for both LOCI- and QM-APHRODITE, these
low precipitation days were redefined as dry days using precipitation threshold, resulting in more
reliable CWD. Finally, although the PT did not correct wet-day frequency, the CWD for the PT-
APHRODITE were lower because tiny precipitation were also corrected.

Mean R10mm during JJAS obtained by the original APHRODITE estimates in the TP, HB, and

FP were 6.7, 31.0, and 47.7 days, respectively. These were similar to those estimated by the bias-
corrected APHRODITE datasets. However, the differences in R20mm were much pronounced.
Mean R20mm in HB and FP for the bias-corrected APHRODITE datasets were close to 19.0 and
26.5 days, respectively, which were approximately 4–5 days higher than those derived from the
original APHRODITE estimates.

Compared with the original APHRODITE estimates, the Rx1d and Rx5d increased greatly after

bias correction. In the HB, the mean Rx1d obtained from the original APHRODITE estimates was
49.5 mm, while those for LS-, LOCI-, PT-, and QM-APHRODITE estimates were 72.4, 90.1, 109.0,
and 103.8 mm, respectively. In addition, the range of Rx1d and Rx5d also increased considerably.
LS was not able to adjust the coefficient of variation, resulting in the lowest Rx1d and Rx5d among
the corrected estimates. Likewise, although precipitation intensity was changed, the Rx1d and Rx5d
for the LOCI-APHRODITE were not as high as those obtained from the two nonlinear corrections,
because it used consistent ratio in its linear transformation.





The differences in SDII between the original and corrected APHRODITE estimates were also
marked. For example, the mean SDII in the FP calculated from the original APHRODITE estimates
was 13.4 mm. After correction, the mean SDII for LOCI- and QM-APHRODITE estimates
increased to 23.4 and 25.1 mm, respectively. These values were much greater than those derived
from LS- and PT-APHRODITE datasets (15.7 and 17.7 mm). The original APHRODITE estimates
are expected to underestimate SDII. Firstly, the original APHRODITE tended to underestimate
precipitation, resulting in very high precipitation in the HB and TP not being fully captured.
Secondly, the original APHRODITE overestimated wet days instead of substantial dry conditions,
which distorted the estimation of precipitation intensity. Larger values of SDII obtained from the
corrected APHRODITE estimates were expected, and the SDII for LOCI- and QM-APHRODITE
were higher because they correct rainfall amount as well as number of rainy days.

**3.2.3   Relative changes in extreme precipitation indices**
The relative changes in extreme precipitation indices during JJAS based on the original and
corrected APHRODITE estimates are shown in Fig. 5. The CWD for LOCI-, PT-, and QM-
APHRODITE were all lower than the original APHRODITE, yielding relative change rates from
−66% to −27%. This indicates bias corrections decreased the number of rainy days except LS. The
variations in R10mm and R20mm illustrated that the corrected APHRODITE identified much more
extreme precipitation events in the TP. The changes in indices varied considerably for different
correction methods, with the change rates of R20mm in the TP for LS-, LOCI-, PT-, and QM-
APHRODITE being 30.4%, 169.2%, 297.1%, and 317.4%, respectively. For Rx1d, Rx5d, and SDII,
the increases in the HB were much pronounced than those in the FP and TP. Except for the LS-





APHRODITE, the increases in Rx1d and Rx5d in the HB were all above 70% for the corrected
APHRODITE estimates. Clearly, topographic variations profoundly influenced the spatial
distribution of precipitation by altering monsoonal flow, resulting in considerable orographic rainfall
on the windward slopes of the Himalayas (Khandu et al., 2017). Insufficient gauge observations in
the Himalayas caused high uncertainty in the heavy precipitation estimates for the original
APHRODITE. After bias adjustment especially those of nonlinear correction, the heterogeneous
orographic effects on extreme precipitation were captured more accurately.

### 3.2.4    Interannual variation of extreme precipitation

To investigate the interannual variation of extreme precipitation for the original and corrected
APHRODITE, the exceedance probabilities of area-averaged Rx5d during JJAS were compared
(Fig. 6). The Rx5d for corrected APHRODITE differ considerably, and the LOCI-, PT-, and QM-
APHRODITE estimated much higher Rx5d than the original APHRODITE and LS-APHRODITE.
In addition, there were greater variability in the Rx5d derived from PT- and QM-APHRODITE. In
particular, heavier Rx5d with low exceedance probabilities obtained by nonlinear corrections
reflected the increasing interannual variation.

### 4    Discussion

Using two linear and two bias nonlinear methods, we corrected APHRODITE estimates during
JJAS in the YBRB to investigate the effects of different approaches on extreme precipitation
analysis. Regardless of the method used, bias correction improved the performance of rainfall
estimates. Nonetheless, extreme precipitation indices were strongly dependent on the bias correction



approach applied.

A primary problem when using gauge-based gridded data sets for extreme precipitation

analysis is the fundamental mismatch between point-based observations and gridded estimates
(Alexander, 2016). In addition, the spatial coverage of rainfall stations is another major source of
uncertainty, particularly where spatial distributions of precipitation are complex (Donat et al., 2013).
There are currently several approaches for bias correction, ranging from simple linear scaling to
more sophisticated nonlinear methods (Teutschbein and Seibert, 2012). Although mean precipitation
corrected by all bias-corrected approaches were similar, their standard deviations and consequent
extreme precipitation indices varied considerably. In the case of linear corrections, both mean and
standard deviation are multiplied by same factor (Leander and Buishand, 2007), resulting in dubious
variations of precipitation. Nonlinear corrections adjust mean and also coefficient of variation
(Teutschbein and Seibert, 2012), yielding more reliable results. In addition, the typical biases of
rainfall products are related to their identification of too many wet days with low-intensity
precipitation. Among the four bias-corrected approaches applied herein, LS and PT make no change
on the number of rainy days, while LOCI and QM use threshold exceedance to match the wet-day
frequency to the observations. Overall, QM corrects most of the statistical characteristics, and
therefore it is expected to perform better in extreme precipitation analysis.

In international river basins, rainfall data are usually not publicly available, and extreme

precipitation analysis may suffer from data restrictions (Nishat and Rahman, 2009; Luo et al., 2019).
Several great international rivers in south Asia, including the Indus, Ganges, and Yarlung
Tsangpo−Brahmaputra, originate from or flow through the Himalayas. Rainfall estimates of
different products varied markedly along the Himalayan front and obtained similar results toward



the adjacent low-relief domains (Andermann et al., 2011). The GHCN-Daily data can be applied to

adjust gauge-based gridded data sets in this region, ensuring these products capture the spatial

distribution and variation of extreme precipitation. However, numerous GHCN-Daily records in

Asia do not contain data from recent years, and the short or incomplete rainfall records limit their

direct applications (Donat et al., 2013). Hence, it would be preferable to add spatial coverage in

data-sparse regions by applying nonpublic datasets.

## 5   Conclusions

Despite increasing use of gridded rainfall products in sparsely gauged river basins, their

application in extreme precipitation analysis is challenging due to considerable biases. This study

made use of four methods to correct the APHRODITE in the YBRB. Their influences on extreme

precipitation indices were compared and assessed. The following conclusions were drawn.

(1) Original APHRODITE tended to underestimate precipitation during JJAS, and bias

correction improved the accuracy of APHRODITE, especially in the HB with complex topography,

highlighting the superiority of corrected APHRODITE.

(2) The extreme precipitation indices calculated from different corrected APHRODITE varied

substantially, depending on correction method and location. Major dissimilarities were induced by

wet-day frequency and standard deviation. Nonlinear correction methods adjust not only mean

precipitation but also coefficient of variation, and QM further corrects probability of wet days,

which is expected to perform better in extreme precipitation analysis.

(3) The deficiency of gauge-based gridded data is mainly attributed to the spatial coverage of

rainfall stations, causing uncertainty to be amplified in extreme precipitation analysis. By correcting

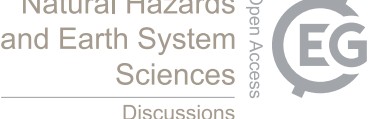

these gauge-based gridded data using complementary observations from denser networks of rainfall
stations, extreme precipitation representation may be greatly improved.

*Data availability*. The co-authors used publicly available data from the Asian Precipitation Highly
Resolved Observational Data Integration Towards Evaluation of Water Resources and the National
Centers for Environmental Information. In addition, rainfall observations in China were obtained
from the National Meteorological Information Center.

*Author contributions*. XL and YL conceived the study, XL and XF carried out bias correction and
extreme precipitation analysis, XL drafted the paper, and all co-authors jointly worked on enriching
and developing the draft.

*Competing interests*. The authors declare that they have no conflict of interest.

*Acknowledgements*. This study was supported by the National Natural Science Foundation of China
(41661144044, 41601026), the National Key R&D Program of China (2016YFA0601601), and the
Science and Technology Planning Project of Yunnan Province, China (2017FB073).

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



**Table 1.** Statistical metrics used in the evaluation of original and corrected APHRODITE estimates.
**Table 2.** Detailed description of extreme precipitation indices.
**Table 3.** Statistical metrics for daily precipitation during JJAS calculated from original and
corrected APHRODITE estimates in the Yarlung Tsangpo-Brahmaputra River Basin (YBRB).





**Table 1.** Statistical metrics used in the evaluation of original and corrected APHRODITE estimates.

| Statistical metric | Equation | Optimal value |
|---|---|---|
| Pearson correlation coefficient ($r$) | $r = \dfrac{\sum_{i=1}^{n}\left(P_{obs,i} - \overline{P_{obs}}\right)\left(P_{APH,i} - \overline{P_{APH}}\right)}{\sqrt{\sum_{i=1}^{n}\left(P_{obs,i} - \overline{P_{obs}}\right)^2}\sqrt{\sum_{i=1}^{n}\left(P_{APH,i} - \overline{P_{APH}}\right)^2}}$ | 1 |
| Percentage bias ($PB$) | $PB = \dfrac{\sum_{i=1}^{n}\left(P_{APH,i} - P_{obs,i}\right)}{\sum_{i=1}^{n}P_{obs,i}} \times 100$ | 0 |
| Mean error ($ME$) | $ME = \dfrac{\sum_{i=1}^{n}\left(P_{APH,i} - P_{obs,i}\right)}{n}$ | 0 |
| Mean absolute error ($MAE$) | $MAE = \dfrac{\sum_{i=1}^{n}\left|P_{APH,i} - P_{obs,i}\right|}{n}$ | 0 |
| Root mean squared error ($RMSE$) | $RMSE = \sqrt{\dfrac{\sum_{i=1}^{n}\left(P_{APH,i} - P_{obs,i}\right)^2}{n}}$ | 0 |

Notation: $n$ means the number of samples; $P_{obs,i}$ and $P_{APH,i}$ refer to the observations and the APHRODITE estimates,
respectively; $\overline{P_{obs}}$ and $\overline{P_{APH}}$ are the mean rain gauge precipitation measurement and the mean APHRODITE
estimate, respectively.





**Table 2.** Detailed description of extreme precipitation indices.

| Index | Descriptive name | Definition | Unit |
|-------|------------------|------------|------|
| CWD | Consecutive wet days | Maximum number of consecutive days with precipitation ≥ 1 mm | days |
| R10mm | Number of heavy precipitation days | Count of days when precipitation ≥ 10 mm during June, July, August, and September (JJAS) | days |
| R20mm | Number of very heavy precipitation days | Count of days when precipitation ≥ 20 mm during JJAS | days |
| Rx1d | Maximum 1-day precipitation amount | Maximum 1-day precipitation | mm |
| Rx5d | Maximum 5-day precipitation amount | Maximum consecutive 5-day precipitation | mm |
| SDII | Simple daily intensity index | Total precipitation during JJAS divided by the number of wet days (when precipitation ≥ 1 mm) | mm/day |






**Table 3.** Statistical metrics for daily precipitation during JJAS calculated from original and corrected APHRODITE estimates in the Yarlung Tsangpo-Brahmaputra River Basin (YBRB).

| Physiographic zone | Correction method | $r$ | PB | ME (mm) | MAE (mm) | RMSE (mm) |
|---|---|---|---|---|---|---|
| TP | Original | 0.80 | −9.4 | −0.3 | 1.7 | 3.4 |
| | Linear scaling | 0.81 | 0.0 | 0.0 | 1.7 | 3.3 |
| | Local intensity scaling | 0.81 | 0.0 | 0.0 | 1.5 | 3.3 |
| | Power transformation | 0.79 | −0.4 | 0.0 | 1.6 | 3.5 |
| | Quantile−quantile mapping | 0.80 | 3.2 | 0.1 | 1.6 | 3.6 |
| HB | Original | 0.81 | −24.2 | −1.6 | 3.6 | 15.9 |
| | Linear scaling | 0.93 | −0.1 | 0.0 | 2.9 | 8.9 |
| | Local intensity scaling | 0.92 | −0.1 | 0.0 | 2.7 | 8.8 |
| | Power transformation | 0.93 | 0.3 | 0.0 | 2.7 | 8.8 |
| | Quantile−quantile mapping | 0.93 | 11.3 | 0.7 | 3.0 | 10.7 |
| FP | Original | 0.81 | −11.4 | −1.5 | 8.0 | 15.5 |
| | Linear scaling | 0.83 | −0.3 | 0.0 | 7.8 | 14.2 |
| | Local intensity scaling | 0.83 | −0.3 | 0.0 | 7.3 | 14.1 |
| | Power transformation | 0.82 | 0.4 | 0.1 | 7.5 | 14.9 |
| | Quantile−quantile mapping | 0.82 | 5.7 | 0.8 | 7.6 | 15.1 |




**Figure 1.** Locations of rainfall stations in the Yarlung Tsangpo-Brahmaputra River Basin (YBRB).
**Figure 2.** Location of Asian Precipitation Highly Resolved Observational Data Integration Towards
Evaluation of Water Resources (APHRODITE) grids over the Tibetan plateau (TP), Himalayan belt
(HB), and floodplains (FP).
**Figure 3.** Spatial distribution of mean maximum 5-day precipitation amount (Rx5d) during June,
July, August, and September (JJAS) in the YBRB based on (a) original APHRODITE, as well as (b)
linear scaling (LS)-APHRODITE, (c) local intensity scaling (LOCI)-APHRODITE, (d) power
transformation (PT)-APHRODITE, and (e) quantile−quantile mapping (QM)-APHRODITE.
**Figure 4.** Box-whisker plot for (a) consecutive wet days (CWD), (b) number of heavy precipitation
days (R10mm), (c) number of very heavy precipitation days (R20mm), (d) maximum 1-day
precipitation amount (Rx1d), (e) Rx5d, and (f) simple daily intensity index (SDII) during JJAS in
the three different physiographic zones (TP, HB, and FP) of YBRB derived from original and
corrected APHRODITE estimates.
**Figure 5.** Relative change rate of (a) CWD, (b) R10mm, (c) R20mm, (d) Rx1d, (e) Rx5d, and (f)
SDII during JJAS for the original and corrected APHRODITE estimates.
**Figure 6.** Exceedance probabilities of area-averaged Rx5d during JJAS for the original and
corrected APHRODITE estimates in the (a) TP, (b) HB, and (c) FP.





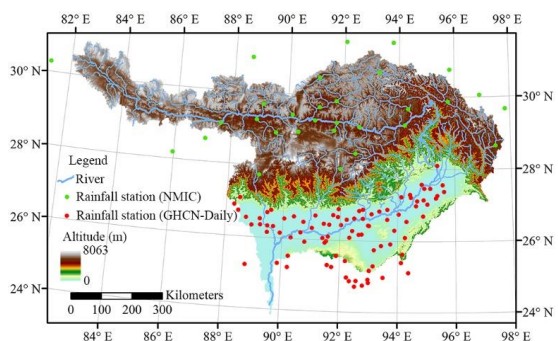


**Figure 1.** Locations of rainfall stations in the Yarlung Tsangpo-Brahmaputra River Basin (YBRB).



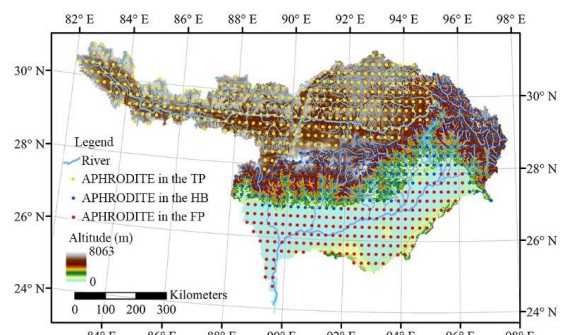


**Figure 2.** Location of Asian Precipitation Highly Resolved Observational Data Integration Towards

Evaluation of Water Resources (APHRODITE) grids over the Tibetan plateau (TP), Himalayan belt
(HB), and floodplains (FP).

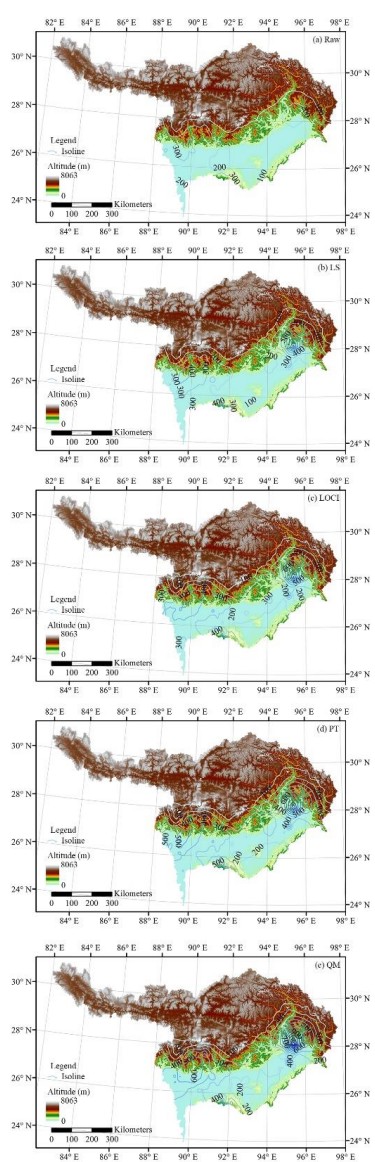


**Figure 3.** Spatial distribution of mean maximum 5-day precipitation amount (Rx5d) during June,

July, August, and September (JJAS) in the YBRB based on (a) original APHRODITE, as well as (b)

linear scaling (LS)-APHRODITE, (c) local intensity scaling (LOCI)-APHRODITE, (d) power

transformation (PT)-APHRODITE, and (e) quantile−quantile mapping (QM)-APHRODITE.

560



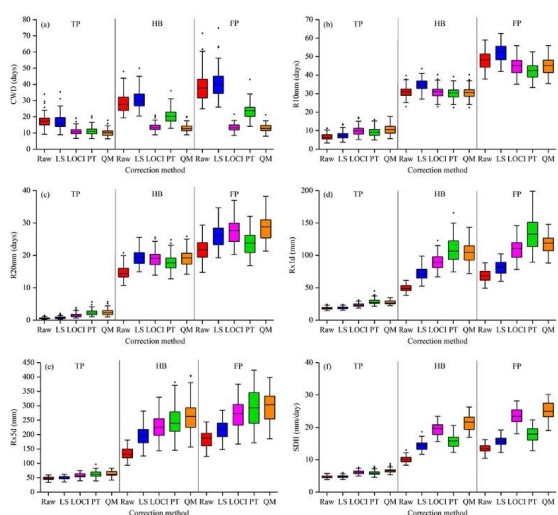

**Figure 4.** Box-whisker plot for (a) consecutive wet days (CWD), (b) number of heavy precipitation

days (R10mm), (c) number of very heavy precipitation days (R20mm), (d) maximum 1-day

precipitation amount (Rx1d), (e) Rx5d, and (f) simple daily intensity index (SDII) during JJAS in

the three different physiographic zones (TP, HB, and FP) of YBRB derived from original and

corrected APHRODITE estimates.

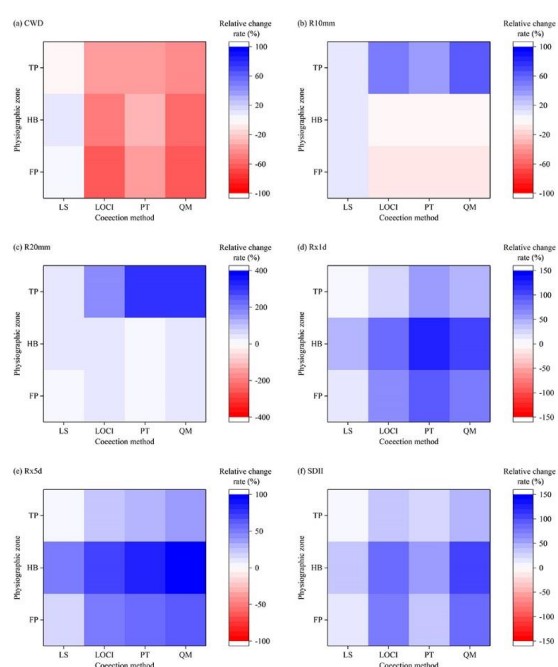

568

**Figure 5.** Relative change rate of (a) CWD, (b) R10mm, (c) R20mm, (d) Rx1d, (e) Rx5d, and (f)

SDII during JJAS for the original and corrected APHRODITE estimates.

571

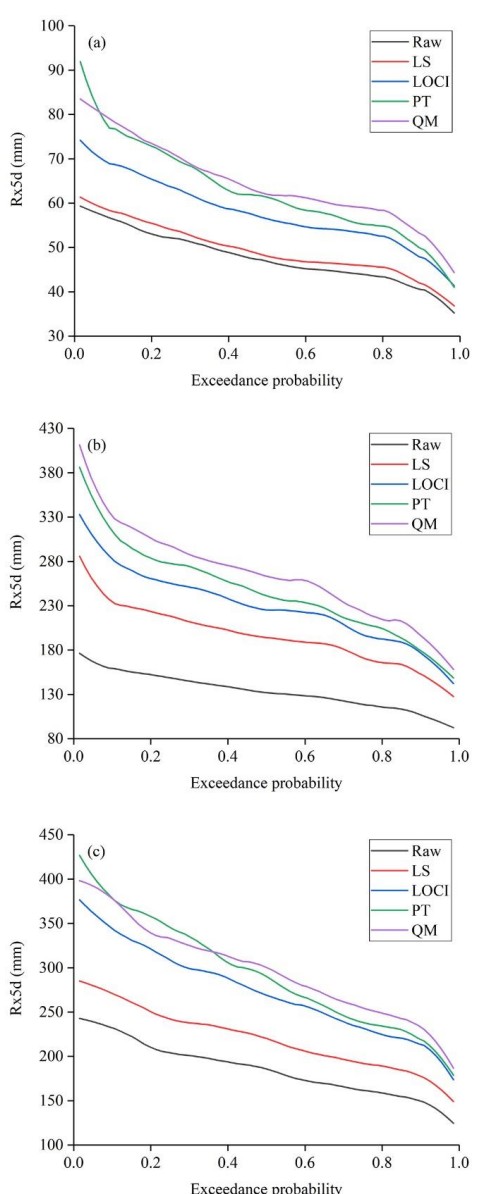

572

**Figure 6.** Exceedance probabilities of area-averaged Rx5d during JJAS for the original and

corrected APHRODITE estimates in the (a) TP, (b) HB, and (c) FP.