# Peer review of "Bias correction of gauge-based gridded product to improve extreme precipitation analysis in the"

_Natural Hazards and Earth System Sciences, 2019_

## Referee Comment (RC1) · Anonymous Referee #1 · 19 Nov 2019

I have carefully read the article: "Bias correction of gauge-based gridded product to improve extreme precipitation analysis in the Yarlung Tsangpo-Brahmaputra River Basin." By Luo et al. While I find that the results of the authors are interesting, I don't quite see how they amount to novel and publishable results as they stand. I should stress that my field of research, in the strictest sense, is bias correction of hydrological data from regional climate models for use in impact model forecasts. So, although I am well informed in matters concerning bias correction of observations, there may be something in the significance of this article that I am not quite understanding.

[Figure]

The authors are using two sources of non-gridded observational data (NMIC and GHCN) to bias correct APHRODITE which is a gridded observational dataset. They use 4 well established Bias Correction (BC) methods. The first two are essentially multiplicative correction factors. They differ in that the second uses a wet-day correction. The third is a variational method, fitted to correct mean and variance, and the last is a parametric Quantile Mapping BC. All these methods are well established, their pros and cons are well studied.

As far as I can see, the authors use all the available non-gridded data to correct the APHRODITE data-set, and then examine the effects of the different BC methods against the very same non-gridded data-set that was used for BCing. This implies that all the comparative results (section: "Evaluation of APHRODITE estimates") are only demonstrative of the mathematical construction of the BC methods and not of any increase in the skill of the corrected APHRODITE data. In simple words, if you bias correct a model to an observation, then, trivially, it looks like that observation. In climate forecasting, one uses past observations and hindcasts to calibrate the BC method and, subsequently, applies the results to bias correct future climate simulations. To validate the BC method one divides the observations into two periods and uses one for correction and one for validation. The studies I have reviewed where observations are bias-corrected, usually divide their observations into two groups as well, one for correction and one for validation, alternatively they sometimes use a leave-one-out cross-validation method. Again, unless I missed something, the comparisons of extreme events indexes between corrected and raw APHRODITE, while insightful, doesn't tell us anything about which one is better since we do not have observations of extreme event statistics from the non-gridded data.

In conclusion, I suggest that the authors extend their work to validate the bias-corrected APHRODITE against observations that were not used in the calibration process and then resubmit their work. Below are line-by-line comments the authors may find useful. Comments:

Line 30 to 32: I do not think the authors have proven this statement: "Bias correction [. . .] greatly improves the performance of extreme precipitation analysis"

Line 36 to 38: I do not see how the results, since they are not cross-validated, help select a bias correction method. Moreover, there are many more bias correction methods available in the literature than those mentioned in this article. See Teutschbein and Seibert 2012 or Cannon et al. 2015

Cannon, A, et al. Bias Correction of GCM Precipitation by Quantile Mapping: How Well Do Methods Preserve Changes in Quantiles and Extremes?, Alex J. Cannon*, Stephen R. Sobie, and Trevor Q. Murdock, Pacific Climate Impacts Consortium, University of Victoria, Victoria, British Columbia, Canada. https://doi.org/10.1175/JCLI-D-14-00754.1

Line 90: As explained above, I do not think the authors "evaluated" as much as "described" their performance.

Line 125: The sentence: "The ratios of rainfall observations. . . " and the sentence after are unclear.

Line 146: I find the indexing not to be exhaustively clear. Is Pobs a station data value? Has the corresponding PAPH been interpolated or vice versa?

Line 153: I know what a wet day correction is but I doubt anyone who does not would understand this sentence.

Line 172: Why show the Gamma density if you are fitting the CDF? Indeed, why write a generic functional form at all? What do the authors mean by "matched"? Is it "fitted"?

Section 2.3.2: I do not see the need for 5 different error measurements.

Line 203 to 204: IDW has serious effects on extreme value distributions. The authors should compare what the distributions look like before and after interpolation.

Section Results: As explained above, results in section 3.1 are unsurprising, while

section 3.2 are not clearly useful to APHRODITE data users.

Line 250 to 251: I do not see how the authors can say this.

Section Conclusions: The authors draw three conclusions and, in the strictest sense, I agree with all of them. This is because the first conclusion is unsurprising while the last two are couched as possibilities instead of results. I refer to language such as: "... is expected to perform better in extreme precipitation analysis" and "extreme precipitation may be greatly improved". While I absolutely agree with these two statements, they are not novel.

---

## Referee Comment (RC2) · Yoann Robin (Referee) · 11 Dec 2019

**1  General comments**

The paper by Luo X. and al propose to compare the performance of four bias correction methods (Linear Scaling, Local Intensity Scaling, Power transformation and Quantile Mapping) of daily precipitations during 1951-2015 over Yarlung Tsangpo-Brahmaputra River Bassin (YBRB). The data to correct comes from the gridded APHRODITE

dataset, and the reference dataset are sparse observations from meteorological stations.

In the first section the dataset and methods are introduced, but it is not clear what time series is matched between model and observations ?  The observations are interpolated ? Aggregated ?

Strangely, the authors can compute the error due to biased correction method between corrected dataset and observations used as reference, but not for extremal indexes. So the final section discussed the influence of bias correction but can not check the quality of the correction itself. Furthermore, a classic way to test of a bias correction is the cross validation step. (cut a least the dataset into calibration and validation period, and swap its), not used here.

In the state of the paper, I can recommend the publication only after majors revisions taking into accounts (at least) of my comments, detailed below.

**2   Specific comments**

**Section 2.2** At the end of this section, you have two dataset, APHRODITE (to correct) and some observations at stations. The stations, considering the Fig. 1 does not match with the grid of APHRODITE, so how do you associate the time series to be correct with the reference time series ?  At the end of this section we need to know exactly what is the biased dataset X matched with the observations Y.

**Section 2.3**

Interactive
comment

[Figure]

- Related to previous questions, I am not sure to understand exactly whats is Pobs and Paph. For example : Paph is a time series of APHRODITE at a grid point and Pobs is the interpolation of observations to correspond to the grid of APHRODITE ? Or you aggregate all data in your three zones TP; HB and FP ?

- For the quantile mapping, how do you fit the Gamma distribution ? MLE ? Moments ? What is the error of the fit ? (I think the error of quantile mapping comes from also from the error of the Gamma model)

**Section 3.2.1** This section is based on the description of Fig. 3, which is not readable, please remove the colormap of topography and increase the size.

**Section 3.2.2 to end of section 3** You compare the original and bias corrected dataset. But without reference, how can you assess an improvment or a degradation by the bias correction method ? You can investigate the effect of bias correction, but not the quality of the correction.

―――――――――――――――――――――

---

## Author Comment (AC1) · 2 Feb 2020

1. I have carefully read the article: "Bias correction of gauge-based gridded product to improve extreme precipitation analysis in the Yarlung Tsangpo-Brahmaputra River Basin." By Luo et al. While I find that the results of the authors are interesting, I don't quite see how they amount to novel and publishable results as they stand. I should stress that my field of research, in the strictest sense, is bias correction of hydrological data from regional climate models for use in impact model forecasts. So, although I am well informed in matters concerning bias correction of observations, there may be

something in the significance of this article that I am not quite understanding. The authors are using two sources of non-gridded observational data (NMIC and GHCN) to bias correct APHRODITE which is a gridded observational dataset. They use 4 well established Bias Correction (BC) methods. The first two are essentially multiplicative correction factors. They differ in that the second uses a wet-day correction. The third is a variational method, fitted to correct mean and variance, and the last is a parametric Quantile Mapping BC. All these methods are well established, their pros and cons are well studied. As far as I can see, the authors use all the available non-gridded data to correct the APHRODITE data-set, and then examine the effects of the different BC methods against the very same non-gridded data-set that was used for BCing. This implies that all the comparative results (section: "Evaluation of APHRODITE estimates") are only demonstrative of the mathematical construction of the BC methods and not of any increase in the skill of the corrected APHRODITE data. In simple words, if you bias correct a model to an observation, then, trivially, it looks like that observation. In climate forecasting, one uses past observations and hindcasts to calibrate the BC method and, subsequently, applies the results to bias correct future climate simulations. To validate the BC method one divides the observations into two periods and uses one for correction and one for validation. The studies I have reviewed where observations are bias-corrected, usually divide their observations into two groups as well, one for correction and one for validation, alternatively they sometimes use a leave one-out cross-validation method. Again, unless I missed something, the comparisons of extreme events indexes between corrected and raw APHRODITE, while insightful, doesn't tell us anything about which one is better since we do not have observations of extreme event statistics from the non-gridded data. In conclusion, I suggest that the authors extend their work to validate the bias-corrected APHRODITE against observations that were not used in the calibration process and then resubmit their work. Below are line-by-line comments the authors may find useful.

Response: Thank you for your comments concerning our manuscript entitled "Bias correction of gauge-based gridded product to improve extreme precipitation analysis in

the Yarlung Tsangpo-Brahmaputra River Basin". Those comments are all valuable and very helpful for revising and improving our paper, as well as the important guiding significance to our researches. We have studied comments carefully and have extended our work and made corrections. As you pointed out, the observations are usually divided into two periods to validate the bias correction, and one is used for correction and the other for validation. This study used non-gridded observational data (NMIC and GHCN) to correct APHRODITE. However, many records collected from GHCN are usually short and incomplete. It is difficult to divide short records into two groups. Alternatively, a leave one-out cross-validation method could be also used to validate bias correction. To improve our manuscript, we have used this method to compare extreme events indexes between corrected and raw APHRODITE. The observations in each one of the rainfall stations was leaved and used to calculate extreme precipitation indices alternately for validation. The observations in all other rainfall stations were used for bias correction and extreme precipitation analysis, and extreme event statistics in the rainfall station for validation were obtained from interpolation and compared with the results calculated from observations. A new figure named "Mean error of extreme precipitation indices for leave one-out cross-validation in the YBRB" was added. By using leave one-out cross-validation, the comparisons of extreme events indexes among raw APHRODITE and different corrected APHRODITE could be more reliable, and QM was proved to be better than the other 3 methods.

2. Line 30 to 32: I do not think the authors have proven this statement: "Bias correction [. . .] greatly improves the performance of extreme precipitation analysis".

Response: After using leave one-out cross-validation, it could be found that bias correction greatly improves the performance of extreme precipitation analysis.

3. Line 36 to 38: I do not see how the results, since they are not cross-validated, help select a bias correction method. Moreover, there are many more bias correction methods available in the literature than those mentioned in this article. See Teutschbein and Seibert 2012 or Cannon et al. 2015 Cannon, A, et al. Bias Correction of GCM Precipitation by Quantile Mapping: How Well Do Methods Preserve Changes in Quantiles and Extremes?, Alex J. Cannon*, Stephen R. Sobie, and Trevor Q. Murdock, Pacific Climate Impacts Consortium, University of Victoria, Victoria, British Columbia, Canada. https://doi.org/10.1175/JCLI-D-14-00754.1

Response: The results has been further cross-validated, which could help select a bias correction method. There are more bias correction methods available than those mentioned in this article. We have added some statements about these methods in the paper.

4. Line 90: As explained above, I do not think the authors "evaluated" as much as "described" their performance.

Response: We have further evaluated the performances of bias corrections using leave one-out cross-validation method.

5. Line 125: The sentence: "The ratios of rainfall observations. . . " and the sentence after are unclear.

Response: We have modified the sentence. The rainfall observations that had undergone quality control were gathered, and the ratios of rainfall observations to the world climatology were calculated and then interpolated for each month.

6. Line 146: I find the indexing not to be exhaustively clear. Is Pobs a station data value? Has the corresponding PAPH been interpolated or vice versa?

Response: We have modified the statement about the variables. Pobs is the observation at rainfall station, while PAPH is APHRODITE estimate at corresponding grid.

7. Line 153: I know what a wet day correction is but I doubt anyone who does not would understand this sentence.

Response: We have modified the statement. Firstly, an adjusted precipitation threshold is determined so that the number of days exceeding this threshold for APHRODITE

estimates matches the number of observed days with rainfall larger than 0 mm.

8. Line 172: Why show the Gamma density if you are fitting the CDF? Indeed, why write a generic functional form at all? What do the authors mean by "matched"? Is it "fitted"?

Response: We have added the formula and explanation of cumulative distribution function besides density function. In this study, generic functional form of Gamma distribution is used. Besides, we have modified statement about quantile–quantile mapping. The cumulative density function (CDF) of the APHRODITE estimates is corrected to agree with that of the observation, and the daily precipitation for APHRODITE estimates is corrected depending on its quantile.

9. Section 2.3.2: I do not see the need for 5 different error measurements.

Response: We have deleted 4 statistical metrics (Pearson correlation coefficient (r), Percentage bias (PB), Mean absolute error (MAE), and Root mean squared error (RMSE)). Mean error (ME) was used in the evaluation of original and corrected APHRODITE estimates.

10. Line 203 to 204: IDW has serious effects on extreme value distributions. The authors should compare what the distributions look like before and after interpolation.

Response: We selected IDW to interpolate extreme precipitation indices due to its deterministic feature. Interpolation methods can be divided into exact interpolation and approximate interpolation. When exact interpolation (such as IDW) is used, interpolation surface moves exactly through each of the points, while the surface of approximate interpolation dose not. In the Yarlung Tsangpo-Brahmaputra River Basin, influenced by complex topography, extreme precipitation vary greatly. The application of IDW could ensure that the interpolated results equal to the values in these grids so that extreme precipitation indices would be more reliable. In addition, the results of leave one-out cross-validation also showed that IDW could be used in the interpolation.

11. Section Results: As explained above, results in section 3.1 are unsurprising, while section 3.2 are not clearly useful to APHRODITE data users.

Response: The results of leave one-out cross-validation were put into section 3.1 to better compare the performance of different bias correction methods. To make the results more useful to APHRODITE data users, some characteristics of APHRODITE estimates were summarized in section 3.2, and the advantage of different bias correction methods were also further analyzed.

12. Line 250 to 251: I do not see how the authors can say this.

Response: We have deleted this sentence.

13. Section Conclusions: The authors draw three conclusions and, in the strictest sense, I agree with all of them. This is because the first conclusion is unsurprising while the last two are couched as possibilities instead of results. I refer to language such as: ". . . is expected to perform better in extreme precipitation analysis" and "extreme precipitation may be greatly improved". While I absolutely agree with these two statements, they are not novel.

Response: The statement of the section of conclusions were modified, and the features of APHRODITE estimates and the advantage of bias correction on extreme precipitation analysis were further summarized. Thanks for all of your suggestions.

**Fig. 1.** Mean error of extreme precipitation indices for leave one-out cross-validation in the YBRB

---

## Author Comment (AC2) · 2 Feb 2020

**1. General comments**

The paper by Luo X. and al propose to compare the performance of four bias correction methods (Linear Scaling, Local Intensity Scaling, Power transformation and Quantile Mapping) of daily precipitations during 1951-2015 over Yarlung Tsangpo-Brahmaputra River Bassin (YBRB). The data to correct comes from the gridded APHRODITE dataset, and the reference dataset are sparse observations from meteorological sta-

tions. In the first section the dataset and methods are introduced, but it is not clear what time series is matched between model and observations? The observations are interpolated? Aggregated? Strangely, the authors can compute the error due to biased correction method between corrected dataset and observations used as reference, but not for extremal indexes. So the final section discussed the influence of bias correction but can not check the quality of the correction itself. Furthermore, a classic way to test of a bias correction is the cross validation step. (cut a least the dataset into calibration and validation period, and swap its), not used here. In the state of the paper, I can recommend the publication only after major revisions taking into accounts (at least) of my comments, detailed below.

Response: Thank you for your comments concerning our manuscript entitled "Bias correction of gauge-based gridded product to improve extreme precipitation analysis in the Yarlung Tsangpo-Brahmaputra River Basin". We have revised the manuscript according to your kind advices and detailed suggestions. We have added statements about the calculation and interpolation of extreme precipitation indices. In the grids distributed with rainfall stations, the parameters of bias corrections were determined using corresponding available rainfall observations. After that, APHRODITE estimates during $1951-2015$ in these grids were corrected, and extreme precipitation indices were then calculated. Finally, spatial interpolation was performed to obtain extreme precipitation indices in other grids with no rainfall station distributed. To improve the study, we have used leave one-out cross-validation method to further discuss the quality of the different bias correction methods and the influence of bias correction on extreme precipitation analysis. The observations in each one of the rainfall stations was leaved and used to calculate extreme precipitation indices alternately for validation. The observations in all other rainfall stations were used for bias correction and extreme precipitation analysis, and extreme event statistics in the rainfall station for validation were obtained from interpolation and compared with the results calculated from observations. A new figure named "Mean error of extreme precipitation indices for leave one-out cross-validation in the YBRB" was added.

**2. Specific comments**

(1) Section 2.2 At the end of this section, you have two dataset, APHRODITE (to correct) and some observations at stations. The stations, considering the Fig. 1 does not match with the grid of APHRODITE, so how do you associate the time series to be correct with the reference time series? At the end of this section we need to know exactly what is the biased dataset X matched with the observations Y.

Response: We have added statements about the bias correction on APHRODITE estimates and calculation of extreme precipitation indices. This study associated the observation at rainfall stations with the APHRODITE estimates according to the location and observation time. In the grids distributed with rainfall stations, the parameters of bias corrections were determined by using corresponding available rainfall observations. APHRODITE estimates during $1951-2015$ in these grids were corrected, and extreme precipitation indices were then calculated. Finally, extreme precipitation indices in other grids with no rainfall station distributed were obtained by spatial interpolation.

(2) Section 2.3 Related to previous questions, I am not sure to understand exactly whats is Pobs and Paph. For example: Paph is a time series of APHRODITE at a grid point and Pobs is the interpolation of observations to correspond to the grid of APHRODITE? Or you aggregate all data in your three zones TP; HB and FP?

Response: We have modified the explanation about the variables. Pobs is the observation at rainfall station, while PAPH is APHRODITE estimate at corresponding grid.

(3) Section 2.3 For the quantile mapping, how do you fit the Gamma distribution? MLE? Moments? What is the error of the fit? (I think the error of quantile mapping comes from also from the error of the Gamma model)

Response: For the quantile mapping, we fit the Gamma distribution by maximum likelihood estimates. Though the Gamma distribution has been proven to be effective in

precipitation analysis, error could be also caused by the Gamma model.

(4) Section 3.2.1 This section is based on the description of Fig. 3, which is not readable, please remove the colormap of topography and increase the size.

Response: To make Fig. 3 more readable, we have removed the topography and increased the size.

(5) Section 3.2.2 to end of section 3 You compare the original and bias corrected dataset. But without reference, how can you assess an improvement or a degradation by the bias correction method? You can investigate the effect of bias correction, but not the quality of the correction.

Response: We have added the analysis on leave one-out cross-validation and modified section 3.2. By using leave one-out cross-validation method, we studied the quality of the different bias correction methods on extreme precipitation analysis. The effects of bias correction were further investigated by comparing the original and corrected APHRODITE estimates. Thanks for all of your suggestions.

[Figure]

**Fig. 1.** Mean error of extreme precipitation indices for leave one-out cross-validation in the YBRB

---

## Author Response (AR1)

**Response to the Reviews on "Bias Correction of Gauge-based Gridded Product to Improve Extreme Precipitation Analysis in the Yarlung Tsangpo-Brahmaputra River Basin"**

**(nhess-2019-327)**

**Responses to Reviewer 1**

1. I have carefully read the article: "Bias correction of gauge-based gridded product to improve extreme precipitation analysis in the Yarlung Tsangpo-Brahmaputra River Basin." By Luo et al. While I find that the results of the authors are interesting, I don't quite see how they amount to novel and publishable results as they stand. I should stress that my field of research, in the strictest sense, is bias correction of hydrological data from regional climate models for use in impact model forecasts. So, although I am well informed in matters concerning bias correction of observations, there may be something in the significance of this article that I am not quite understanding.

The authors are using two sources of non-gridded observational data (NMIC and GHCN) to bias correct APHRODITE which is a gridded observational dataset. They use 4 well established Bias Correction (BC) methods. The first two are essentially multiplicative correction factors. They differ in that the second uses a wet-day correction. The third is a variational method, fitted to correct mean and variance, and the last is a parametric Quantile Mapping BC. All these methods are well established, their pros and cons are well studied.

As far as I can see, the authors use all the available non-gridded data to correct the APHRODITE data-set, and then examine the effects of the different BC methods against the very same non-gridded data-set that was used for BCing. This implies that all the comparative results (section: "Evaluation of APHRODITE estimates") are only demonstrative of the mathematical construction of the BC methods and not of any increase in the skill of the corrected APHRODITE data. In simple words, if you bias correct a model to an observation, then, trivially, it looks like that observation. In climate forecasting, one uses past observations and hindcasts to calibrate the BC method and, subsequently, applies the results to bias correct future climate simulations. To validate the BC method one divides the observations into two periods and uses one for correction and one for validation. The studies I have reviewed where observations are bias-corrected, usually divide their observations into two groups as well, one for correction and one for validation, alternatively they sometimes use a leave one-out cross-validation method. Again, unless I missed something, the comparisons of extreme events indexes between corrected and raw APHRODITE, while insightful, doesn't tell us anything about which one is better since we do not have observations of extreme event statistics from the non-gridded data.

In conclusion, I suggest that the authors extend their work to validate the bias-corrected APHRODITE against observations that were not used in the calibration process and then resubmit their work. Below are line-by-line comments the authors may find useful.

Response: Thank you for your comments concerning our manuscript entitled "Bias correction of gauge-based gridded product to improve extreme precipitation analysis in the Yarlung Tsangpo-Brahmaputra River Basin". Those comments are all valuable and very helpful for revising and improving our paper, as well as the important guiding significance to our researches. We have studied comments carefully and have extended our work and made corrections.

As you pointed out, the observations are usually divided into two periods to validate the bias correction, and one is applied for correction and the other for validation. However, GHCN-Daily records used in this study are mostly short and incomplete, and it is difficult to divide these short records into two groups. Alternatively, a leave one-out cross-validation method could also be used to validate bias correction. To improve our manuscript, we have used this method to compare extreme events indexes between corrected and original APHRODITE estimates. The observations in each one of the rainfall stations were leaved and applied to calculate extreme precipitation indices alternately for validation. The observations in all other rainfall stations were used for bias correction and extreme precipitation analysis, and extreme precipitation indices in the rainfall station for validation were obtained from interpolation and then compared with the results calculated from observations. A new figure named "Mean error of extreme precipitation indices for leave one-out cross-validation in the YBRB" was added. By using leave one-out cross-validation, the comparisons of extreme events indexes among raw APHRODITE estimates and different corrected APHRODITE estimates could be more reliable, and QM was proved to perform better than the other 3 methods.

2. Line 30 to 32: I do not think the authors have proven this statement: "Bias correction [. . .] greatly improves the performance of extreme precipitation analysis".

Response: After using leave one-out cross-validation, it could be found that bias correction greatly improves the performance of extreme precipitation analysis.

3. Line 36 to 38: I do not see how the results, since they are not cross-validated, help select a bias correction method. Moreover, there are many more bias correction methods available in the literature than those mentioned in this article. See Teutschbein and Seibert 2012 or Cannon et al. 2015

Cannon, A, et al. Bias Correction of GCM Precipitation by Quantile Mapping: How Well Do Methods Preserve Changes in Quantiles and Extremes?, Alex J. Cannon*, Stephen R. Sobie, and Trevor Q. Murdock, Pacific Climate Impacts Consortium, University of Victoria, Victoria, British Columbia, Canada. https://doi.org/10.1175/JCLI-D-14-00754.1

Response: The results has been further cross-validated, which could help select a bias correction method.

In the paper of "Bias correction of regional climate model simulations for hydrological climate-change impact studies: Review and evaluation of different methods" written by Teutschbein and Seibert, 6 bias correction methods were used to adjust RCM simulations: (1) linear scaling, (2) local intensity scaling, (3) power transformation, (4) variance scaling, (5) distribution transfer, and (6) delta-change approach. Among them, variance scaling was only applied to correct temperature, while delta-change approach only correct precipitation for the future scenario. Except them, the other 4 methods were all used in this study.

In the paper of "Bias Correction of GCM Precipitation by Quantile Mapping: How Well Do Methods Preserve Changes in Quantiles and Extremes", 3 bias correction methods were used to adjust RCM simulations: (1) quantile mapping (QM), (2) quantile delta mapping (QDM), and detrended quantile mapping (DQM). QM has been used in this study. The QDM preserves model-projected relative changes in quantiles, while at the same time correcting systematic biases in quantiles of a modeled series with respect to observed values. While DQM incorporates additional information about the climate model outputs in the projected period. As QDM and DQM are used to correct systematic distributional biases in precipitation outputs from climate models, we have not used them to correct APHRODITE estimates.

4. Line 90: As explained above, I do not think the authors "evaluated" as much as "described" their performance.

Response: We have further evaluated the performances of bias corrections using leave one-out cross-validation method.

5. Line 125: The sentence: "The ratios of rainfall observations. . . " and the sentence after are unclear.

Response: We have modified the sentence. The rainfall observations that had undergone quality control were gathered, and the ratios of rainfall observations to the world climatology were calculated and then interpolated for each month.

6. Line 146: I find the indexing not to be exhaustively clear. Is Pobs a station data value? Has the corresponding PAPH been interpolated or vice versa?

Response: We have modified the statement about the variables. Pobs is the daily precipitation at rainfall station, and PAPH is the daily precipitation of the APHRODITE estimate at corresponding grid.

7. Line 153: I know what a wet day correction is but I doubt anyone who does not would understand this sentence.

Response: We have modified the statement. Firstly, an adjusted precipitation threshold is determined so that the number of days exceeding this threshold for APHRODITE estimates matches that of observed days with precipitation larger than 0 mm.

8. Line 172: Why show the Gamma density if you are fitting the CDF? Indeed, why write a generic functional form at all? What do the authors mean by "matched"? Is it "fitted"?

Response: The CDF of Gamma distribution is not integrabel, and we did not show it. The Gamma distribution has been proven to be effective in precipitation analysis, and this form was used in other papers. Besides, we have modified statement about quantile–quantile mapping. The cumulative density function (CDF) of the APHRODITE estimates is adjusted to agree with that of the observation, and the daily precipitation for APHRODITE estimates is corrected depending on its quantile.

9. Section 2.3.2: I do not see the need for 5 different error measurements.

Response: As leave one-out cross-validation method was used, we have deleted the section of evaluation of original and corrected APHRODITE estimates. By comparing with the extreme precipitation indices obtained from observation, mean error (ME) was used to evaluate the performances of original and corrected APHRODITE estimates on extreme precipitation analysis.

10. Line 203 to 204: IDW has serious effects on extreme value distributions. The authors should compare what the distributions look like before and after interpolation.

Response: We selected IDW to interpolate extreme precipitation indices due to its deterministic feature. Interpolation methods can be divided into exact interpolation and approximate interpolation. When exact interpolation (such as IDW) is used, interpolation surface moves exactly through each of the points, while the surface of approximate interpolation dose not. In the Yarlung Tsangpo-Brahmaputra River Basin, influenced by complex topography, extreme precipitation vary greatly. The application of IDW could ensure that the interpolated results equal to the values in the sample points so that extreme precipitation indices would be more reliable. We have used leave one-out cross-validation to compare observations and the results obtained by interpolation.

11. Section Results: As explained above, results in section 3.1 are unsurprising, while section 3.2 are not clearly useful to APHRODITE data users.

Response: Leave one-out cross-validation was used to better compare the performance of different bias correction methods. To make the results more useful to APHRODITE data users, some characteristics of APHRODITE estimates were summarized, and the advantage of different bias correction methods were also further analyzed.

12. Line 250 to 251: I do not see how the authors can say this.

Response: We have deleted this sentence.

13. Section Conclusions: The authors draw three conclusions and, in the strictest sense, I agree with all of them. This is because the first conclusion is unsurprising while the last two are couched as possibilities instead of results. I refer to language such as: ". . . is expected to perform better in extreme precipitation analysis" and "extreme precipitation may be greatly improved". While I

absolutely agree with these two statements, they are not novel.

Response: The statement of conclusions were modified. Thanks for all of your suggestions.

**Responses to Reviewer 2**

1. General comments

The paper by Luo X. and al propose to compare the performance of four bias correction methods (Linear Scaling, Local Intensity Scaling, Power transformation and Quantile Mapping) of daily precipitations during 1951-2015 over Yarlung Tsangpo-Brahmaputra River Bassin (YBRB). The data to correct comes from the gridded APHRODITE dataset, and the reference dataset are sparse observations from meteorological stations.

In the first section the dataset and methods are introduced, but it is not clear what time series is matched between model and observations? The observations are interpolated? Aggregated?

Strangely, the authors can compute the error due to biased correction method between corrected dataset and observations used as reference, but not for extremal indexes. So the final section discussed the influence of bias correction but can not check the quality of the correction itself. Furthermore, a classic way to test of a bias correction is the cross validation step. (cut a least the dataset into calibration and validation period, and swap its), not used here.

In the state of the paper, I can recommend the publication only after major revisions taking into accounts (at least) of my comments, detailed below.

Response: Thank you for your comments concerning our manuscript entitled "Bias correction of gauge-based gridded product to improve extreme precipitation analysis in the Yarlung Tsangpo-Brahmaputra River Basin". We have revised the manuscript according to your kind advices and detailed suggestions.

We have added statements about the calculation and interpolation of extreme precipitation indices. This study associated the observation at rainfall stations with the APHRODITE estimates according to the location and observation time. In the grids distributed with rainfall stations, the parameters of bias corrections were determined using corresponding available rainfall observations. APHRODITE estimates during 1951−2015 in these grids were corrected by 4 bias correction methods, respectively. After that, extreme precipitation indices for corrected APHRODITE estimates in the grids distributed with rainfall stations were calculated. To obtain extreme precipitation indices in other grids with no rainfall station distributed, spatial interpolation was performed using inverse distance weighted (IDW) interpolation.

To improve the study, we have used a leave one-out cross-validation method to further discuss the quality of the different bias correction methods and the influence of bias correction on extreme precipitation analysis. The observations in each one of the rainfall stations was leaved and used to calculate extreme precipitation indices alternately for validation. The observations in all other rainfall stations were used for bias correction and extreme precipitation analysis, and extreme precipitation indices in the rainfall station for validation were obtained from interpolation and compared with the results calculated from observations. A new figure named "Mean error of extreme precipitation indices for leave one-out cross-validation in the YBRB" was added.

2. Specific comments (1) Section 2.2 At the end of this section, you have two dataset, APHRODITE (to correct) and some observations at stations. The stations, considering the Fig. 1 does not match with the grid of APHRODITE, so how do you associate the time series to be correct with the reference time series? At the end of this section we need to know exactly what is the biased dataset X matched with the observations Y.

Response: We have added statements about the bias correction on APHRODITE estimates and calculation of extreme precipitation indices. This study associated the observation at rainfall stations with the APHRODITE estimates according to the location and observation time. In the grids distributed with rainfall stations, the parameters of bias corrections were determined using corresponding available rainfall observations. After that, APHRODITE estimates during 1951−2015 in these grids were corrected by 4 bias correction methods, respectively.

(2) Section 2.3 Related to previous questions, I am not sure to understand exactly what is Pobs and Paph. For example: Paph is a time series of APHRODITE at a grid point and Pobs is the interpolation of observations to correspond to the grid of APHRODITE? Or you aggregate all data in your three zones TP; HB and FP?

Response: We have modified the explanation about the variables. Pobs is the observation at rainfall station, while PAPH is APHRODITE estimate at corresponding grid.

(3) Section 2.3 For the quantile mapping, how do you fit the Gamma distribution? MLE? Moments? What is the error of the fit? (I think the error of quantile mapping comes from also from the error of the Gamma model)

Response: For the quantile mapping, we fit the Gamma distribution by maximum likelihood estimates. Though the Gamma distribution has been proven to be effective in precipitation analysis, error could be caused by the Gamma model.

(4) Section 3.2.1 This section is based on the description of Fig. 3, which is not readable, please remove the colormap of topography and increase the size.

Response: As the topography in this figure is used to analyze the impacts of the Himalayas on the spatial distribution of extreme precipitation, we have not removed the colormap of topography. To make this figure more readable, we have modified the transparency of the colormap of topography, and the colors of isolines were also changed. Besides, we have increased the size of this figure.

(5) Section 3.2.2 to end of section 3 You compare the original and bias corrected dataset. But without reference, how can you assess an improvement or a degradation by the bias correction method? You can investigate the effect of bias correction, but not the quality of the correction.

Response: We have added the analysis on leave one-out cross-validation and modified section 3.2. By using leave one-out cross-validation, we studied the quality of different bias correction on extreme precipitation analysis. Thanks for all of your suggestions.

[revised manuscript text omitted]

---

## Referee Report (RR1)

**Review of "Bias correction of gauge-based gridded product to improve extreme precipitation analysis in the Yarlung Tsangpo-Brahmaputra River Basin"**

*Authors : Xian Luo, Xuemei Fan, Yungang Li, and Xuan Ji*

**Summary**

The paper by Luo X. and al proposes to compare the performance of four bias correction methods (Linear Scaling, Local Intensity Scaling, Power transformation and Quantile Mapping) of daily precipitations during 1951-2015 over Yarlung Tsangpo-Brahmaputra River Bassin (YBRB). The data to correct comes from the gridded APHRODITE dataset, and the reference dataset are sparse observations from meteorological stations. The performance of bias correction methods is evaluated with a spatial leave one-out cross validation method: one station is removed, and an IDW interpolation between others bias corrected grid point is applied to build them.

**General comments**

Having already participated in the first review round, I am glad to see that my comments have been taken into account. The main addition is the leave one-out cross validation. Contrary to the usual practice, instead of a cross-validation in time (the dataset is split into two time periods, which makes it possible to check the stationarity of the probability distribution in the context of this paper), the authors propose a leave one-out cross validation by removing alternatively one station from observational data.

I'm not convinced this approach can validate the quality of bias correction. Eventually, this method can be applied if the bias correction method is multivariate (dependence structure between grid points is also corrected, with methods as MBCn, R2D2 or dOTC, see Cannon, Vrac and Robin). In this context, it is not the bias correction method that is tested, but the interpolation method.

I'm sorry, but I can not understand why the authors can not split the dataset into two time periods even if it means removing stations when no data is available for the sub time period. Furthermore, cutting does not require the sub-periods to be continuous. The heart of this paper is the improvement due to a bias correction, and the prerequisite is to verify that the method reproduces well the distribution of the observations. The only element in this sense is the figure 6, which shows a better coherence with the topography.

I can not recommend publication without a clear proof of improvement compared to observations, and it is not the case (all the figures show a modification compared to APHRODITE, but do not show if it is closed to observations, or more realistic).

**Specific comments**

**Lines 188-189**

The sentence "according to the location and observation time." is not really clear. It is my understanding that the authors correct only the grid points of APHRODITE that contain a time series of observations (and an interpolation is used for others), but I am note sure. Please clarify.

**Lines 207-209**

The sentence "To obtain extreme precipitation indices in other grids with no rainfall station distributed, spatial interpolation was performed using inverse distance weighted (IDW) interpolation" is slightly confusing. You perform the interpolation between the bias corrected dataset or between the extreme precipitation indices computed from the bias corrected dataset ?

**Technical comments**

**Figure 4**

In the x-label : correction instead of "coeeection".

**References**

Cannon, A. J.: Multivariate quantile mapping bias correction: an N-dimensional probability density function transform for climate model simulations of multiple variables, Climate Dynamics, 50, 31–49, 2018.

Vrac, M.: Multivariate bias adjustment of high-dimensional climate simulations: the Rank Resampling for Distributions and Dependences (R2 D2 ) bias correction, Hydrology and Earth System Sciences, 22, 3175–3196, 2018.

Robin, Y., Vrac, M., Naveau, P., and Yiou, P.: Multivariate stochastic bias corrections with optimal transport, Hydrology and Earth System Sciences, 23, 773–786, 2019.

---

## Referee Report (RR2)

**Review of "Bias correction of gauge-based gridded product to improve extreme precipitation analysis in the Yarlung Tsangpo-Brahmaputra River Basin"**

*Authors : Xian Luo, Xuemei Fan, Yungang Li, and Xuan Ji*

**Summary**

The paper by Luo X. and al proposes to compare the performance of four bias correction methods (Linear Scaling, Local Intensity Scaling, Power transformation and Quantile Mapping) of daily precipitations during 1951-2015 over Yarlung Tsangpo-Brahmaputra River Bassin (YBRB). The data to correct comes from the gridded APHRODITE dataset, and the reference dataset are sparse observations from meteorological stations. The performance of bias correction methods is evaluated with a time cross validation method.

**General comments**

I am pleased to see that the authors have incorporated a time cross validation to asses the capacity of bias correction methods tested to reduce the bias. I think the paper contains the necessary scientific material, and I just have two suggestions.

First, the section 2.3.3 is a little too short, and would benefit from being better explained. You cut the dataset in half, use two-thirds of the first half to calculate the parameters of a bias correction method, and then correct the second half? Finally you calculate the RMSE between the corrected second half and the observations?

Second, I think that section 3.2.1 should be at the beginning of section 3. It seems more logical to me to first check the cross-validation and then discuss the corrections themselves.

**Specific comments**

**Technical comments**

---

## Author Response (AR2)

**Response to the Reviews on "Bias Correction of Gauge-based Gridded Product to Improve Extreme Precipitation Analysis in the Yarlung Tsangpo-Brahmaputra River Basin"**

**(nhess-2019-327)**

**Responses to Editor**

The paper has been well improved, and this is acknowledged by the referees. However, I agree with reviewer 1 about the insufficient proof of the efficiency of the correction with the chosen protocol. One possible way of improvement could be, for the left out station, to compare the indices obtained from the series reconstructed by interpolation before and after correction, so that the role of the correction can be more clearly separated from that of the spatial interpolation. This may not be a major revision, except if no real improvement is found after correction.

Response: Thank you for the comments concerning our manuscript entitled "Bias Correction of Gauge-based Gridded Product to Improve Extreme Precipitation Analysis in the Yarlung Tsangpo-Brahmaputra River Basin" (ID: nhess-2019-327). Those comments are all valuable and very helpful. To improve the paper, we have used a cross-validation in time instead of a spatial leave one-out cross validation to evaluate the performance of different bias correction methods. In addition, we have made some other corrections.

**Responses to Reviewer 1**

The authors have implemented the simpler of the two suggested cross-validation options. The results from the new analysis support the authors' earlier claims. I now find that this work is fit for publication. I would, however, suggest to the authors to have their manuscript professionally edited. Slight changes in syntax will improve legibility.

Response: Thank you again for your valuable and helpful comments concerning our manuscript. We have further made some changes in syntax to improve the paper.

**Responses to Reviewer 2**

1. Summary

The paper by Luo X. and al proposes to compare the performance of four bias correction methods (Linear Scaling, Local Intensity Scaling, Power transformation and Quantile Mapping) of daily

precipitations during 1951-2015 over Yarlung Tsangpo-Brahmaputra River Bassin (YBRB). The data to correct comes from the gridded APHRODITE dataset, and the reference dataset are sparse observations from meteorological stations. The performance of bias correction methods is evaluated with a spatial leave one-out cross validation method: one station is removed, and an IDW interpolation between others bias corrected grid point is applied to build them.

Response: Thank you again for your valuable and helpful comments concerning our manuscript. We have carefully studied comments and made corrections, and the performance of bias correction methods were evaluated by a cross-validation in time instead of a spatial leave one-out cross validation.

2. General comments

Having already participated in the first review round, I am glad to see that my comments have been taken into account. The main addition is the leave one-out cross validation. Contrary to the usual practice, instead of a cross-validation in time (the dataset is split into two time periods, which makes it possible to check the stationarity of the probability distribution in the context of this paper), the authors propose a leave one-out cross validation by removing alternatively one station from observational data.

I'm not convinced this approach can validate the quality of bias correction. Eventually, this method can be applied if the bias correction method is multivariate (dependence structure between grid points is also corrected, with methods as MBCn, R2D2 or dOTC, see Cannon, Vrac and Robin). In this context, it is not the bias correction method that is tested, but the interpolation method.

I'm sorry, but I can not understand why the authors can not split the dataset into two time periods even if it means removing stations when no data is available for the sub time period. Furthermore, cutting does not require the sub-periods to be continuous. The heart of this paper is the improvement due to a bias correction, and the prerequisite is to verify that the method reproduces well the distribution of the observations. The only element in this sense is the figure 6, which shows a better coherence with the topography.

I can not recommend publication without a clear proof of improvement compared to observations, and it is not the case (all the figures show a modification compared to APHRODITE, but do not show if it is closed to observations, or more realistic).

Response: To better compare the extreme precipitation indices calculated from corrected APHRODITE estimates with those from observations, we have applied a cross-validation in time instead of a spatial leave one-out cross validation. At each rainfall station, the observations were divided into two groups. Two third of the rainfall records were applied to calculate the parameters of bias correction, and then APHRODITE estimates were corrected. Making use of the remaining rainfall observations, the mean error (*ME*) of the extreme precipitation indices for corrected APHRODITE estimates were calculated to evaluate the performance of different bias correction methods. Fig. 5 was redrawn to show the *ME* of extreme precipitation indices for validation.

The results showed that bias correction greatly improved the performance of extreme precipitation analysis, and local intensity scaling (LOCI) and quantile−quantile mapping (QM) performed better than linear scaling (LS) and power transformation (PT).

3. Specific comments

(1) Lines 188-189

The sentence "according to the location and observation time." is not really clear. It is my understanding that the authors correct only the grid points of APHRODITE that contain a time series of observations (and an interpolation is used for others), but I am note sure. Please clarify.

Response: We have modified the sentence as "This study corrected the grids of the APHRODITE estimates that contained time series of observations".

(2) Lines 207-209

The sentence "To obtain extreme precipitation indices in other grids with no rainfall station distributed, spatial interpolation was performed using inverse distance weighted (IDW) interpolation" is slightly confusing. You perform the interpolation between the bias corrected dataset or between the extreme precipitation indices computed from the bias corrected dataset?

Response: We have modified the sentence as "To obtain extreme precipitation indices in other grids, inverse distance weighted (IDW) interpolation for extreme precipitation indices were performed".

4. Technical comments

Figure 4

In the x-label: correction instead of "coeeection".

Response: We have replaced "coeeection" with "correction" in the x-label in Figure 4. Thanks for all of your suggestions.

[revised manuscript text omitted]

---

## Author Response (AR3)

**Response to the Reviews on "Bias Correction of Gauge-based Gridded Product to Improve Extreme Precipitation Analysis in the Yarlung Tsangpo-Brahmaputra River Basin"**

**(nhess-2019-327)**

**Responses to Editor**

Thanks for your work, I'm following the recommendation of the reviewer.

Response: Thank you again for the works concerning our manuscript entitled "Bias Correction of Gauge-based Gridded Product to Improve Extreme Precipitation Analysis in the Yarlung Tsangpo-Brahmaputra River Basin" (ID: nhess-2019-327). We have made some corrections on this paper.

**Responses to Reviewer**

I am pleased to see that the authors have incorporated a time cross validation to assess the capacity of bias correction methods tested to reduce the bias. I think the paper contains the necessary scientific material, and I just have two suggestions.

First, the section 2.3.3 is a little too short, and would benefit from being better explained. You cut the dataset in half, use two-thirds of the first half to calculate the parameters of a bias correction method, and then correct the second half? Finally you calculate the RMSE between the corrected second half and the observations?

Second, I think that section 3.2.1 should be at the beginning of section 3. It seems more logical to me to first check the cross-validation and then discuss the corrections themselves.

Response: Thank you for your valuable and helpful comments concerning our manuscript. We have carefully studied comments and made corrections.

(1) We have modified the section 2.3.3 to better explain the validation on bias correction. Cross-validation was applied to evaluate the performance of four bias correction methods. At each rainfall station, the observations were divided into two groups. Two-thirds of the rainfall records were applied to calculate the parameters of LS, LOCI, PT, and QM, respectively. Making use of these parameters, the APHRODITE estimates were then corrected. The mean error ($ME$) between the extreme precipitation indices obtained from the corrected APHRODITE estimates and those obtained from remaining one-third of the rainfall observations were calculated to evaluate the performance of different bias correction methods.

(2) We have moved the section of evaluation of extreme precipitation indices to the beginning of section 3 so that the results seem more logical.

Thanks for all of your suggestions.

[revised manuscript text omitted]